# Evolutionary advantages of turning points in human cooperative behaviour

Daniele Vilone[1,2]*, John Realpe-Gómez[3,4], Giulia Andrighetto[1,5,6]

**1** LABSS (Laboratory of Agent Based Social Simulation), Institute of Cognitive Science and Technology, National Research Council (CNR), Rome, Italy, **2** Grupo Interdisciplinar de Sistemas Complejos (GISC), Departamento de Matemáticas, Universidad Carlos III de Madrid, Leganés, Spain, **3** Laboratory for Research in Complex Systems, San Francisco, California, United States of America, **4** ICTP South American Institute for Fundamental Research, Instituto de Física Teórica, Universidade Estadual Paulista, São Paulo, Brazil, **5** Mälardalen University, Vasteras, Sweden, **6** Institute for Future Studies, Stockholm, Sweden

\* daniele.vilone@gmail.com

**Data Availability Statement:** This paper is a theoretical and analytical work, with simulation results presented to support the conclusions of the paper. No original experimental data have been utilized. The algorithm of the simulations is

## Abstract

Cooperation is crucial to overcome some of the most pressing social challenges of our times, such as the spreading of infectious diseases, corruption and environmental conservation. Yet, how cooperation emerges and persists is still a puzzle for social scientists. Since human cooperation is individually costly, cooperative attitudes should have been eliminated by natural selection in favour of selfishness. Yet, cooperation is common in human societies, so there must be some features which make it evolutionarily advantageous. Using a cognitive inspired model of human cooperation, recent work Realpe-Gómez (2018) has reported signatures of criticality in human cooperative groups. Theoretical evidence suggests that being poised at a critical point provides evolutionary advantages to groups by enhancing responsiveness of these systems to external attacks. After showing that signatures of criticality can be detected in human cooperative groups composed by Moody Conditional Cooperators, in this work we show that being poised close to a turning point enhances the fitness and make individuals more resistant to invasions by free riders.

## 1 Introduction

Cooperation is a backbone of our existence as an extraordinary social species. It enables the joint pursuit of political objectives and the more prosaic foundations of everyday life [1, 2], especially in group interactions [3, 4]. Yet how human cooperation is achieved and persists is still a significant puzzle in science [5–9]. Since human cooperation is individually costly, on first assessment, cooperative attitudes should have been eliminated by natural selection in favour of selfishness. Yet, cooperation is common in human societies, so it must have features which make it evolutionarily advantageous.

A recent work [10], whose main results are summarized in the Appendix of this paper, has reported that groups composed of humans facing a social dilemma are posed near a critical regime: traditionally, the concept of criticality arises in statistical mechanics in the study of phase transitions, and identifies an equilibrium configuration of a system poised at the

described precisely in the paper in Subsection 2.3. The algorithm can be used by any author to replicate the study, properly citing the original work.

**Funding:** For G. A. and D. V., this work was partially supported by the Knut and Wallenberg Grant "How do human norms form and change?" (2016.0167), the Horizon 2020 Framework Programme Project PROTON "Modelling the Processes leading to Organised crime and Terrorist Networks" (No.: 699824), by the Eranet FLAG-ERA Call 2016 Project FuturICT 2.0 – "Large scale experiments and simulations for the second generation of FuturICT" (CUP: B84E16000780005), and the Italian Research Project of National Relevance (PRIN) "14ALL: One for all, all for one. Reputational mechanisms for aggression, revenge, and forgiveness in intergroup relationships". J. R.-G. would like to thank FAPESP grant 2016/01343-7 for funding my visit to ICTP-SAIFR from 20-27 January 2019 where part of this work was done.

**Competing interests:** The authors have declared that no competing interests exist.

boundary between a disordered and an ordered phase (as for instance liquid and solid, ferromagnetic and paramagnetic, and so on) [11–14]. Evidence of critical behaviour has been detected also in animal societies, such as flock of birds [15], ant groups [16], or communities of macaques [17], but to our knowledge [10] is the first to attest signatures of criticality in human cooperative systems too. Theoretical evidence suggests that being poised at a critical point provides functional advantages to a social system [11], which when is far from criticality is either not responsive enough, favouring maladaptive behaviours, or too uncoordinated with its members behaving independently of each other. In both extremes the system as a whole is not very responsive to external changes, while around a critical point it is strongly correlated and highly sensitive to these changes. In particular, reference [11] connects theoretically social and biological systems with criticality, showing that the latter is the evolutionary stable outcome of a group of individuals who continuously tune and adjust their behaviour to the behaviour of others in their attempt to cope with complex and heterogeneous environments.

The aim of this paper is to test the effects of being poised close to a turning point on the fitness of the group and its ability to resist to external attacks. In particular, we focus here on a laboratory experiment described in [18], different from the one investigated in [10], and show that when playing a Prisoner's Dilemma Game, groups composed by Moody Conditional Cooperators (MCCs), a behavioural strategy according to which players choose whether to cooperate or not on the basis of their previous action and the actions of their neighbors, poise close to a turning point. We then prove that MCCs have their fitness enhanced and are more resistant to invasions by free riders when poised close to this point. With respect to [10], in this work we not only investigate whether signatures of criticality can be detected in human groups, but we also test its effect on human cooperative behaviour (and it differs from [16] where the effect of criticality is studied in groups of ants). Finally, we also show that, despite such turning point can not be considered critical in proper terms, it is endowed with features in many ways similar to the ones associated with criticality. To this aim, we conceived a game-theoretical model with a double time-scale dynamics: first, a short-time scale dynamics based on decision-making rules at the level of single games repeated a finite number of rounds, corresponding to that of the experiment we analyze [18] and able to reproduce its results. Secondly, a long-time scale evolutionary dynamics, which allows the advantages of being close to a transition emerge.

The rest of this paper is organized as follows. Section 2 provides a description of the laboratory experiment we study [18] and introduces the short- and long-time dynamics we analyze. Section 3 presents the results of numerical simulations showing that groups of MCC agents playing Prisoner's Dilemma Game (PDG) poise through evolution near a turning point, where they are more resistant to invasion by free riders. The last section summarises our main conclusions. In the the Appendix we provide further technical details.

## 2 Model definitions

### 2.1 Structure of the Prisoner's Dilemma Game studied

Our model follows accurately the experimental set-up by [18], where human subjects strategically interact repeatedly with their $K_{max}$ neighbours on a two-dimensional lattice (while in [19] analysed in [10] the players are posed on a scale-free network). Similarly to [19], agents interact pairwise among themselves according to the so-called Weak Prisoner's Dilemma Game (wPDG), whose payoff matrix is given in Table 1 below. While in the classical PDG, cooperation is always costly with respect to defection, in the wPDG a cooperator and a defector receive the same payoff against a defector. In this game, defection is not a risk dominant option, which enhances the possibility that cooperation emerges [20, 21]. Nevertheless, defection

**Table 1. Payoff matrix for a generic 2 × 2 game.** In the experiments [18, 19] studied here, the values of the payoffs are meant in units of Euro cents; this is called a *weak* PDG (wPDG) because the payoff is always zero when the opponent defects, so agents are not directly punished for cooperating when their peers defect.

|   | C | D |
|---|---|---|
| C | (7,7) | (0,10) |
| D | (10,0) | (0,0) |

results in a better payoff than cooperation when the opponent cooperates, thus making this setting relevant to study the emergence of cooperation.

At each round, every subject plays a PDG with her neighbours in the network and is rewarded with the overall payoff obtained in her interactions with each of her neighbours. The gained payoff is defined as the player's fitness, and depends on both her action and those of her neighbours', calculated on the basis of the matrix given in Table 1. Here, as in [18], the network is a square lattice with a Moore neighbourhood, i.e. $K = 8$ (for further details, see [18]).

## 2.2 Types of players and results of the laboratory experiment

Here we summarize the main results of the laboratory experiment reported in [18] that we analyze: (i) the cooperation level starts from a given point (30% in the first realization and 60% in the second one), and progressively decreases until reaching a seemingly steady state of about 20%; (ii) changing the structure of the network does not significantly affect cooperation as long as the number of neighbours is kept fixed; (iii) players do not take into account the earnings of their neighbours when deciding to cooperate during the game; (iv) three types of players can be identified according to their strategy: "absolute cooperators" ($< 5\%$), i.e., players that always cooperate, free riders or "absolute defectors" ($\simeq 30\%$), players that always defect, and Moody Conditional Cooperators (MCCs) [18, 22]. The rule of behaviour of MCC players is the following:

**After defection**: If the agent has defected in the previous round, in the next one she will cooperate with probability

$$\Pi_{D \to C} = q; \tag{1}$$

**After cooperation**: If the agent has previously cooperated, in the next round she will cooperate with probability

$$\Pi_{C \to C} = \min(1, pK + r), \tag{2}$$

where $K$ is the number of nearest neighbours who have cooperated, and $p$, $q$ and $r$ two model parameters. The min function guarantees that $\Pi_{C \to C} \leq 1$.

We highlight the role of the parameter $p$, which is the more meaningful. Indeed, it is the multiplying factor of $K$, that is, it tunes the driving force of the cooperating neighbours: for $p \to 0^+$ the cooperating neighbours do not influence the agent's behaviour at all, whilst in the limit $p \to 1^-$ such influence is the biggest possible.

Based on these experimental observations, in our simulations we consider three type of agents: absolute cooperators, absolute defectors, and MCCs. Following Eqs (1) and (2), the MCC rule is completely specified by the values of three non-negative parameters, $p$, $q$, $r$. The empirical values reported in the experiment that interests us fluctuate due to the diverse

conditions, but are in the ranges:

$$p \sim 0.08 \div 0.09; \quad q \sim 0.15 \div 0.22; \quad r \sim 0.35 \div 0.38 \ ,$$

(see data of 'exp. 2' in Table 1 of [18]). Therefore, for simplicity we assume for such parameters the following values:

$$p \simeq 0.085; \quad q \simeq 0.2; \quad r \simeq 0.4. \tag{3}$$

## 2.3 Simulations

Starting from the empirical results obtained in [18, 19, 23–25], in this work we consider populations made up of the three types of agents introduced above (absolute cooperators, absolute defectors and MCCs), which play a weak PDG (see Sec. 2.1 and Table 1) on a square lattice with coordination number $K_{max} = 8$. Absolute cooperators and defectors always cooperate or defect, respectively, whilst MCCs at each round decide how to act according to probabilities which depend on their own and their neighbours' previous actions.

Such populations follow two types of dynamics at two different time scales. First, agents play for a given number of rounds (in our simulations we set 100 rounds) behaving according to their type, that is, adapting their actions but keeping their strategy fixed during the entire game. In particular, the behaviour of an agent is not affected by her neighbours' payoffs, consistent with recent experimental findings [18, 19, 22, 25]. Second, at an evolutionary time scale, agents with the most successful strategy, or fitness produce more offspring, allowing the best strategies to propagate over time. Here we refer to fitness as the average payoff accumulated by a player after a given number of rounds $T$, according to the payoff matrix given in Table 1. Consistent with standard evolutionary dynamics, the update of the strategies does take into account the (average) payoffs obtained by other agents [26].

We set this double-scale dynamics in order to make the population evolve and test the behaviour of each type of agent according to their interactions with others. More precisely, the dynamics at long scales is useful to make the best strategies naturally survive at the expense of the worse ones. This will allow us to get an evolutionary rationale of the behaviours observed in the lab. In what follows we discuss in more detail how we implement these two dynamics.

**2.3.1 Non-evolutionary adaptive dynamics within a game.** During the game players may change their actions (cooperate/defect), but they keep their strategy fixed. That is, after fixing the MCC parameters $p$, $q$ and $r$, at each round agents play simultaneously with their neighbours, deciding whether to cooperate or defect according to the rules given in Subsec. 2.2.

To explore how the agents' collective behaviour depends on the parameters of the model, we first study a population composed only of MCC agents. In order to single out the transition more effectively, we keep the parameters $q$ and $r$ fixed and close to the experimental values, $q = 0.2$ and $r = 0.4$ [18], and study the cooperation level reached at the steady state of the dynamics as a function of the parameter $p$. Afterwards, we carry out the simulations with more realistic conditions, using the proportions of MCC players, absolute cooperators, and absolute defectors observed in the laboratory experiment.

To have reliable statistics, we consider a system of $N = 32 \times 32 = 1024$ agents and average over two thousand independent realizations. Each realization starts from completely random initial conditions and is run until a steady state is reached. In contrast, the laboratory experiment [18] considers two independent sessions (plus a control) of $N_r = 13 \times 13 = 169$ subjects and $50 \div 60$ rounds, which may not be enough to reach completely a steady state (assuming

that the real system can reach one). All the other settings, i.e., the network structure, the payoff matrix and the rules of the game coincide with those of the laboratory experiment (see Sec. 2).

**2.3.2 Evolutionary dynamics of strategies over many games.** Here, we describe the evolutionary dynamics used in our simulations. The dynamic rules we employ do not necessarily faithfully represent human actual evolutionary dynamics. Our aim is not to explain how the *Homo sapiens* has evolved up to the behaviour observed in the experiments under study, but to provide some theoretical insights on why in these experiments it is convenient for the individuals to stay as close as possible to a turning point. Nevertheless, (see Sec. 3.1), some of the main features we observe with the simple dynamics we used should also be observed in more realistic dynamics.

The evolutionary algorithm we study is the following:

**Interaction stage**: Agents play a game repeatedly during a number of $T$ rounds; here we select $T = 100$.

**Update stage**: At the end of the interaction stage, each agent selects a neighbour at random and checks her payoff accumulated during the previous interaction stage. If her neighbour's accumulated payoff is strictly larger than her own, the agent imitates her neighbour's current action (*i.e.*, cooperation or defection) and strategy (*i.e.*, absolute cooperator, absolute defector, or MCC player).

**New generation**: After the update stage, the agents reset their fitness and start a new interaction stage.

The rule described in the update stage is similar to that of replicator dynamics [27]. We also used replicator dynamics but it was too slow and the system did not reach the steady state in a reasonable time. This rule takes into account the neighbours' payoffs, which is apparently against the experimental observations reported in [18, 25]. Yet, this evolutionary dynamic operates over many generations, i.e., over many games played, while the experimental findings mentioned apply at the scale of a single game, i.e., played over a fixed number of rounds. Indeed, we let the agents accumulate their payoff during the interaction stage so that when they have to update them, the best strategy have indisputably emerged.

## 3 Results

In this section, we first present a theoretical analysis of the MCC strategy. Later, we present the results of the non-evolutionary simulations. Finally, we summarize the outcomes of the evolutionary simulations.

### 3.1 General analysis of the Moody Conditional Cooperation strategy

**3.1.1 Theoretical analysis.** To better understand the results below, we include a general analysis of Eqs (1) and (2). As this analysis is independent of the dynamics used, we expect that some of the main features we observe in the simulations can be common to other more realistic dynamics as well.

Since $p, r > 0$, the maximum value $\Pi^{\max}_{C \to C}$ of the left hand side of Eq (2) is obtained when $K = K_{max} = 8$. For a fixed value of $r$, the parameter $p$ can be classified in three regimes as follows:

1. $p < \dfrac{1 - r}{K_{\max}}$ so that $\Pi^{\max}_{C \to C} < 1$, i.e., an agent has a finite probability to defect even when all her neighbours cooperated: this may decrease the global fitness of the system (that is, the sum of the payoffs earned by all the players), as it is likely to foster defection by other agents.

2. $p \geq \dfrac{1-r}{K_{\max}-1}$, i.e., an agent cooperates with 100% probability even if at the previous round one neighbour (or more, if $p$ is large enough) had defected; this may render the subject vulnerable to cheating.

3. $\dfrac{1-r}{K_{\max}} \leq p < \dfrac{1-r}{K_{\max}-1}$, i.e., an agent cooperates with 100% probability if all her neighbours has cooperated, and defects with a finite probability if at least one neighbour have defected. In this case, an agent can in principle punish free riders by applying a sort of soft tit-for-tat.

When $K_{max} = 8$, the condition in item 3 above becomes $0.075 \lesssim p \lesssim 0.086$, which largely constrains the value of $p$. We can see that the experimental value for $p$ (see Eq (3)) is consistent with this constraint. It results that, in the laboratory experiment [18], the parameters characterizing the MCC agents are such that $pK_{\max} + r \simeq 1$, so $\Pi_{C \to C}^{\max} \simeq 1$. Setting $p^* = (1-r)/K_{max}$, this indicates that when trying to be as cooperative as possible, but in a way that allows them not to be exploited by defectors (that is, for $p \to p^*$, see item 2 above), MCC agents reach a turning point (see Sec. 3): in Subsec. 3.4, we will provide evidence that being poised close to this point enables the system to be resistant to invasions by free riders.

## 3.2 Non-evolutionary adaptive game dynamics simulations

We now discuss the results of computer simulations conducted to assess the effect of different values of the MCC parameters on the social dynamics of the system. In particular, we analyze the final state reached by the system as a function of the model parameters initially assigned, and compare the outcomes with the observed behaviour of the population in the experiment with humans reported in [18]. This analysis may allow us to better understand what may be the advantages for the population parameters to be close to the empirical ones. We present a set of simulations reproducing the main conditions ($K_{max}$, lattice structure, payoff matrix, initial conditions) of the experiment presented in [18], we leave only the system size free, in order to study the features of the model depending on it. The results are reported in Fig 1, where the final cooperation level reached by the system is shown as a function of the parameter $p$.

First, it is possible to notice an abrupt change at $p = 0.09 \simeq p^*$, where the final cooperation rate suddenly raises to 1: if the system is composed only of MCC players, when approaching $p^*$ they suddenly tend to act as absolute cooperators and always cooperate. As it can be expected, the presence of other types of players smoothens the transition, but the effect is still clearly discernible. The actual value of $p$ measured in the experiments is close to this point.

Second, results show that, when the population is distributed according to the mix of MCC players, absolute cooperators, and absolute defectors, at $p = p^*$ observed in the experiment (red line), the final cooperation level reached in the simulations turns out to be very close to the one observed in the experiments with humans [18] than to other values of $p$, that is, slightly larger than 20%.

## 3.3 Evolutionary dynamics simulations

In order to better understand the experimental results and the effect on the system of being poised nearby $p^*$, we consider the outcomes of the evolutionary simulations defined in Subsection 2.3.2.

In Fig 2, we show the average cooperation rate and the density of MCC agents reached in the final state by the system for sizes $N = 225$ and $N = 1024$, respectively, as functions of the parameter $p$. As it is easy to see, for $p = p^* \simeq 0.09$ MCC agents comprises near all the population and achieve near full cooperation. In contrast, for $p < p^*$, MCCs and absolute defectors

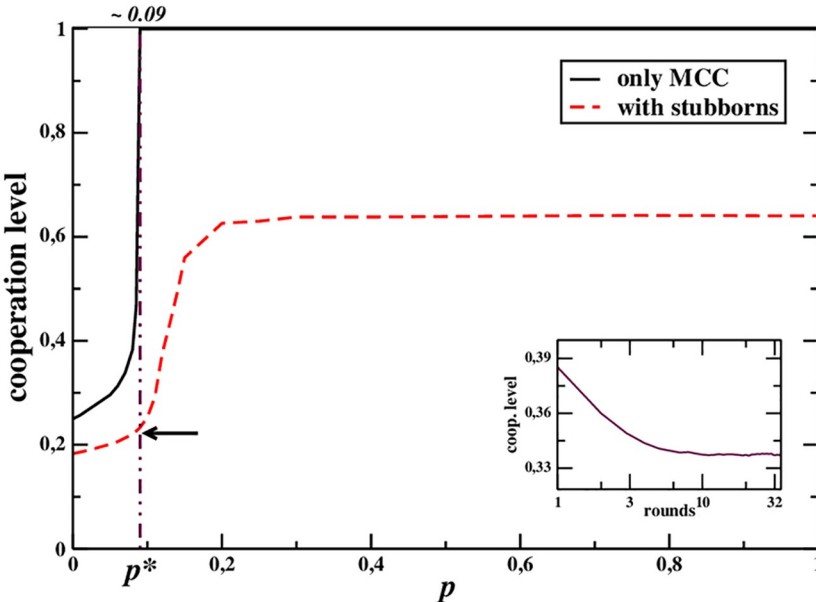

**Fig 1. Long-term cooperation level reached by a population of agents interacting on a square lattice playing a PDG as described in [18].** The population is composed either exclusively of MCC agents (black line), or of a mixture of MCC players and "stubborns", that is, absolute cooperators and absolute defectors, in a proportion equal to that reported in [18] (about 65%, 5%, and 30%, respectively). The results are illustrated as a function of $p$, being the remaining parameters fixed to the experimental values $q = 0.2$, $r = 0.4$; the arrow points at the experimental outcome, showing that the system of agents is very close to $p^*$. **Inset**: Time behaviour (*i.e.*, as game rounds go by) of the cooperation level for a system with $p \simeq p^*$ (empirical value), which is qualitatively very similar to the experimental one (see Fig 1 of Ref. [18]).

coexist, resulting in a lower cooperation rate. This situation arises because MCC agents react to free riders by defecting themselves. Finally, for $p > p^*$, MCCs can easily be exploited by defectors who can then comprise the majority of the population. We also show that absolute cooperators are always quickly wiped out by selection and vanish before the dynamics reaches the steady state (see the Appendix for details), since, in this case, they have a non-zero probability to cooperate also when interacting with free riders.

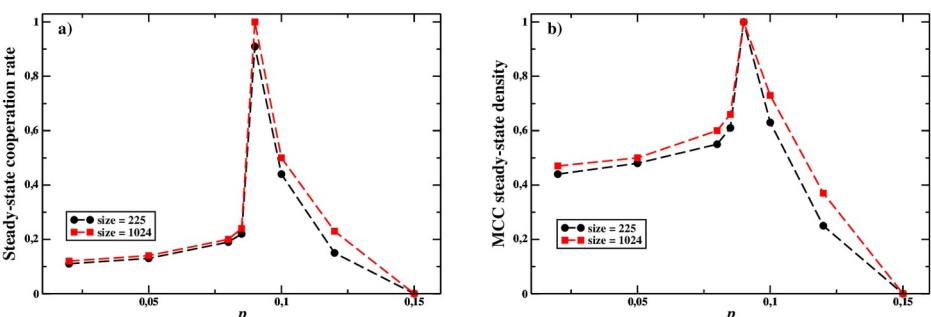

**Fig 2.** a) Steady state values for cooperation level, and b) MCC density for populations of agents interacting on a square lattice, with 8 nearest neighbours per individual, according to a weak Prisoner Dilemma game (see Table 1). We performed simulations also for size $N = 2500$ (not shown), but the results were essentially the same as those for $N = 1024$.

### 3.4 Footprints of criticality at the turning point

We now provide evidence that the critical phenomenon observed around the point $p^* \approx 0.09$ is connected to the phenomenon related in [10] and shows interesting properties of criticality introduced in [11]. Since the value of $p^*$ is close to the experimental value $p \approx 0.085$ reported in [18], this suggests that the human experimental group is near the turning point.

Figs 1 and 2 suggest that at $p = p^*$ cooperative behaviours invade completely the whole system. That is, the transition takes place when the largest cluster of (cooperating) MCCs coincides with the entire system. So, in order to describe better this phenomenon, we resort to the correlation length, $\xi$, which measures the spatial memory of the system (in practice, two agents can influence each other if separated at most by a distance $\xi$) and diverges at least linearly in critical phenomena [28]. We can estimate the correlation length by means of the linear size of the largest connected clusters of cooperators [29]:

$$\xi \approx \max\{\ell_C : \ell_C \text{ linear size of clusters of connected cooperator}\}. \tag{4}$$

Fig 3 shows how the quantity $\xi/L$ depends on the linear system size $L$ for different values of $p$ near $p^*$. We can see that, apart from the case $p = p^*$, it tends to vanish as the system size increases. At $p = p^*$, instead, for large enough systems, the largest connected cluster of cooperators is about the same size as the system, $\xi \sim L$. So, a system large enough—i.e., a system with about $30 \times 30$ agents according to Fig 3—cooperation percolates through the system only around $p \approx p^*$.

How can we interpret this result? Consider a population composed only of MCC agents: as shown in Fig 1, the overall fitness is maximized for $p \geq p^*$ (see Subsec. 3.1). Therefore, on the basis of fitness alone, with fixed $q = 0.2$ and $r = 0.4$, MCC players could indifferently distribute their $p$ value along the interval $[p^*, 1]$. To understand why $p^*$ is effectively the selected value, we have to take into account the outcomes of the evolutionary simulations, which help us to shed light on the benefit of being near $p^*$ for MCC players.

The results shown in Fig 2 can be easily understood as follows. First, for $p < p^*$, the MCC players do not cooperate much but they can nevertheless avoid exploitation and partially

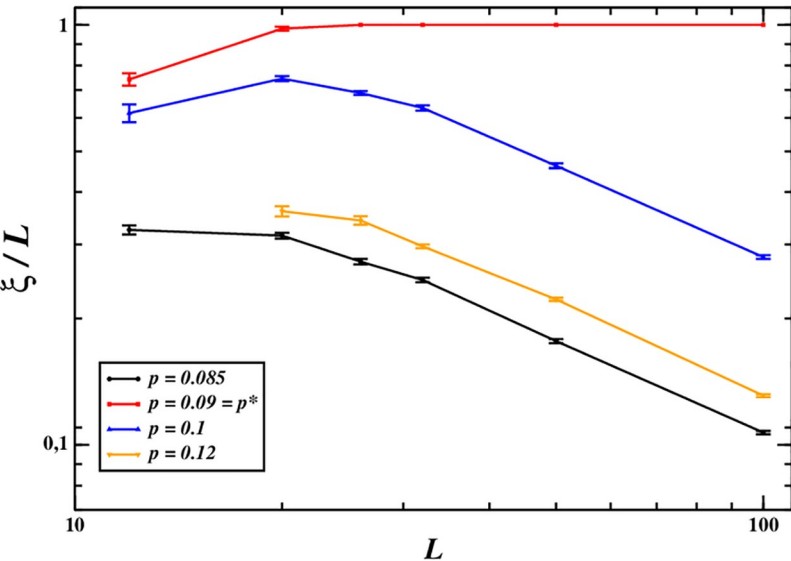

**Fig 3. Behaviour of the ratio $\xi/L$ as a function of the linear size of the system for different values of $p$ near the transition point:** Only at $p^*$ it rapidly reaches 1, while for $p \neq p^*$ it tends to vanish in the thermodynamic limit.

survive. So, the cooperation rate is relatively low due to the coexistence of defectors with MCC players, who cooperate from time to time. However, in this case MCC players are not responsive enough to the cooperation of their MCC peers and end up with a lower level of cooperation than possible. Secondly, for $p > p^*$, MCC agents tend to cooperate too frequently, even when their neighbours are pure defectors. In this case MCC players are not responsive enough to defection. Therefore, MCC agents can be exploited by free riders and become extinct, leaving a population of only defectors. Finally, when $p \approx p^*$, MCC players can eliminate most defectors yielding a cooperation rate very close, or equal, to 1. In this case, MCC agents are responsive enough to both cooperation and defection. In a sense, for $p \approx p^*$ MCC agents optimally respond to both cooperation and defection by their peers in a strategy analogous to the tit-for-tat strategy in pair interactions, and this allows the development of long range correlations poising the system nearby the turning point. Indeed, Fig 3 shows that $\xi$ grows in proportion to the size of the system, $L$, around $p = p^*$, while for $p$ far from $p^*$ it grows sublinearly. More precisely, at $p = p^*$ MCC players invade the population, therefore their largest cluster corresponds to the whole system. It is important to highlight that the presence of absolute defectors—i.e., free riders—is the key factor inducing MCC players to poise themselves near $p^*$: actually, staying close to the critical point can be viewed as the optimum response to the attempt of exploitation by free riders. The behaviour of the correlation length $\xi$ recalls what happens in second-order phase transitions, and we interpret it as a footprint of criticality. Nevertheless, it is not enough to declare the observed phenomenon as a full manifestation of criticality: for instance, the clusters of cooperating MCCs near to $p^*$ do not show scale-invariance, as reported in Fig 5. Therefore, it may be considered a sort of *quasi-critical* phenomenon.

It is important to notice, though, that there are other sources of resistance to free riders. Indeed, a system of MCC agents located on a fully-connected network rather than on a square lattice, displays the same phenomena of Fig 1 (black solid line) when only the non-evolutionary dynamics is considered, as could be shown by simulations or a mean field analysis (results not shown). However, unlike the case of a square lattice considered here, once (deterministic) evolutionary dynamics is considered, defectors take over the system. The reason is that in a fully-connected graph, MCC players cannot form clusters in which cooperators can be isolated from free riders and will therefore always be subject to exploitation by defectors. So, in line with the literature on cooperation [30], here clustering is what allows MCC to survive. On the other hand, the properties of criticality detected in the emergence of a giant cluster of cooperators, allows MCC to thrive by driving defectors to extinction. Nevertheless, if we consider the stochastic evolutionary dynamics associated with finite populations, it is still possible that a small fraction of defectors would become extinct due to drift. Whether criticality can enhance the effects of drift for finite populations is left for future work.

At $p = p^*$ the MCC population is evolutionarily stable so, it is resistant to invasions by free riders. To illustrate this, Fig 4 shows the temporal dynamics of the density of MCCs in a system initially populated only by MCCs which, at a given time, is invaded by a 5% of absolute defectors. As we can see, the invasion is completely absorbed at the turning point, only partially absorbed for $p < p^*$, whilst for $p > p^*$ the invaders end up invading the system completely and the MCC players become extinct. This finding is coherent with [11], where it has been proven in general terms that criticality helps a biological system be also stable.

## 3.5 Spatial distribution of agents

To better understand the mechanisms at work in the phenomenology described up to now, it is useful to consider the spatial distribution of agents and strategies adopted at the metastable state—which coincides with the final state when the system size approaches infinity. In Fig 5

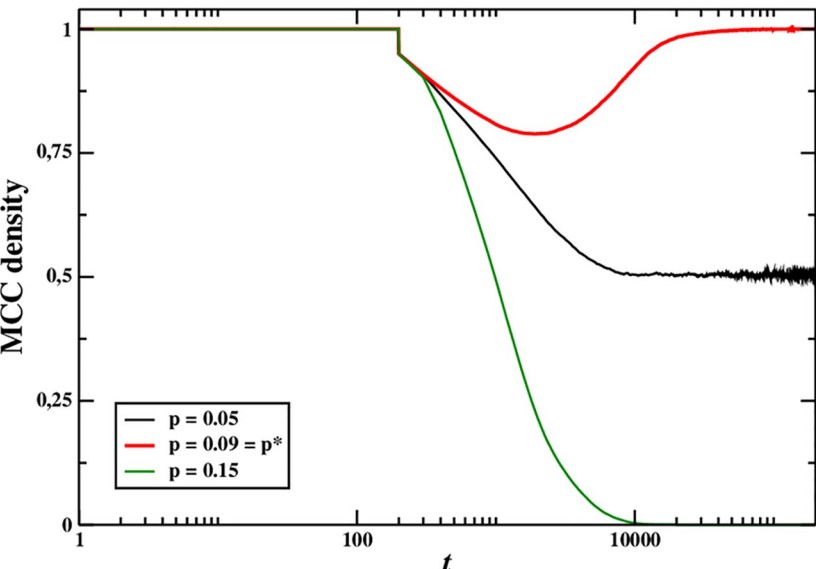

**Fig 4. Time evolution of the MCC density for a population of size $N = 1024$, initially made up of MCC agents only.**
After 200 iterations, 5% of absolute defectors are injected into the population. MCC agents are able to get rid of most free riders for $p = 0.09 = p^*$, but not for $p = 0.15 > p^*$ nor $p = 0.05 < p^*$.

we show the final distribution of agents (a) and strategies (b) for a system of size $101 \times 101$ (in this case we used a larger lattice to make its properties clearer). As we can see in Fig 5a, absolute defectors (black) are relatively few and confined in small clusters that appear to be connected to each other by percolating-like filaments. However, Fig 5b indicates that despite the small presence of defectors, the most adopted action is defection even by the majority of MCC players. This result is because MCC agents directly linked with a cluster of absolute defectors have a high probability to defect in their turn, so that also the MCC agents connected to them are likely to defect: for an MCC agent to be safe from free-riding it is necessary to be far away from absolute defectors. Therefore, only few MCC agents are really free to cooperate. As a consequence, a relatively small percentage of absolute defectors is enough to make the level of cooperation to diminish, and that is why only for $p = p^*$, or very close to it, cooperation level can be high.

Naturally, a numerical and/or theoretical analysis of the model in higher dimensions could shed more light on the nature of this turning point and its relationship with proper critical

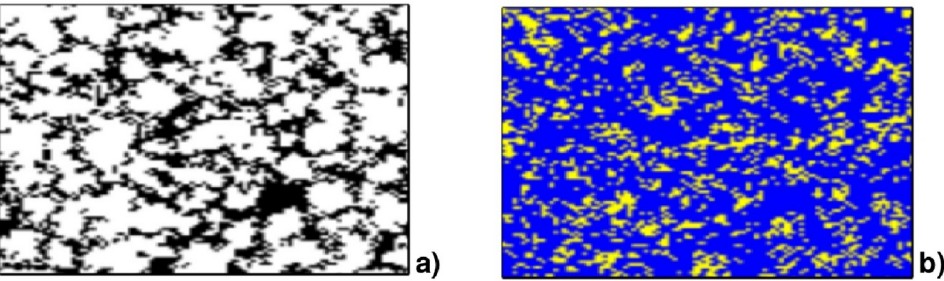

**Fig 5.** Spatial distribution of a) agent types (black: absolute defectors, white: MCCs—absolute cooperators extinct) and b) actions (blue: defection, yellow: cooperation) in the metastable state for a $101 \times 101$ system with $p = 0.085$.

phenomena. In this paper we limited the study to the experimental configuration (that is, a two-dimensional system with $K = 8$), leaving such a systematic study of the transition for future works.

## 4 Conclusions

The simulations and theoretical results presented in this paper, based on the experimental results reported in [18], provide additional evidence that signatures of criticality can be identified in human cooperative groups composed by Moody Conditional Cooperators with the effect of enhancing their fitness and making them more resistant to the invasions by free riders. Our results show that being poised near a turning point provides evolutionary advantages to the population: indeed, MCC players can thrive because they can cooperate when all their neighbours cooperate, and defect when interacting with defectors, thus potentially inducing the latter ones to cooperate. More precisely, when the probability to cooperate again after cooperation (given by $p$) is smaller than $p^*$, MCC players do not always cooperate, even when surrounded by cooperators, which can negatively impact their fitness in the long term. For values of $p$ larger than $p^*$, instead, MCC players cannot avoid being exploited by free riders, because they cooperate also when interacting with some defectors. The property of this transition in fostering cooperation recalls criticality because of the scarce effect of the network's details: that is, even though a structure is necessary to allow the formation of connected clusters of (cooperating) MCCs isolated from free riders, the details of the topology are not important. Although the transition at $p^*$ can not be considered as critical in classical terms, it has important features in common with critical phenomena: the sudden change in the outcome of the dynamics (Fig 1), and the maximization of the global fitness of the whole population (Fig 2), the divergence of the correlation length (Fig 3), due to an optimal response in the interactions among individuals, as described in [11]. In line with previous theoretical work [11], our model shows that when poised near a phase transition, human cooperative groups can more effectively resist invasions by free riders. This unexplored feature of human cooperative systems may help explain why cooperation has evolved despite its costs. More specifically, MCC players increase cooperation because the creation and survival of clusters of cooperators are largely fostered at $p^*$ (see also in the Appendix). As soon as a defector enters the system, her neighbours reduce their tendency to cooperate with her, thus lowering the defector's fitness. In this way, being at the transition enhances the system to reach stability and, as a consequence, the ability to resist invaders.

Up to now, the only experiments with human subjects which have shown signs of criticality are the ones described in references [18] and [19], but it would be certainly interesting to find similar instances in other experiments and generalize our analysis to different and, when possible, more realistic situations. Indeed, if the role of criticality is confirmed in human collective behaviour, fields beyond biology as sociology, psychology, history, and many others in humanities, will have a new, valuable tool to understand the dynamics of society. Therefore, a next step of this kind of research should be to find more compelling empirical manifestations of criticality in human behaviour: new laboratory experiments aimed to find proper phenomena of criticality in this field will be considered for the forthcoming future.

## Appendix: Linking the macroscopic dynamics with cognitive mechanisms

In this section, we connect the MCC macroscopic dynamics studied above with a more microscopic modelling framework, based on empirically-sound cognitive assumptions, that combines individual and norm-based motivations, with model-free and model based learning

mechanisms. The first empirical evidence of criticality in cooperative human groups was identified in [10]. In [10] a population of agents is considered. Such agents have internal mechanisms which allow them to learn and adopt the best strategy to maximize their fitness. This internal dynamic is modelled using the so-called EWAN algorithm. EWAN builds on the Experience-Weighted Attraction (EWA) algorithm introduced in [31] and it is extended to allow agents to recognize, reason and comply with social norms. This integration has been done to account for growing evidence that when choosing whether to cooperate, humans do not always act in order to maximize their personal payoffs, but they also care about behaving in line with the social norms shared by group, namely informal and shared behavioural rules that, unlike legal norms, are not codified but are learnt through social interaction. Those rules prescribe what individuals ought or ought not to do, and whose violation is often enforced through informal punishment, such as ostracism, gossip or dishonour for the transgressor [32–36]. Social norms provide information about how members within a certain group will behave and more impor-tantly about how they are dictated to behave. This ability is modelled via a modified pay-off matrix that includes not only the individual economic gains, as usually done in classical game theory work but also the gains obtained when complying with the social norm. Even though the agents are all of the same type, initially in the same state, and the learning algorithm is rather general, the system reaches a final active configuration with players behaving as MCCs that are poised at a critical point, reproducing results in close agreement with the ones observed empirically [18].

As discussed in [10] and summarized below, under the assumptions of slow adaptation and absence of network reciprocity (experimental finding (ii) mentioned in the Sec. 2.2), it is possible to obtain an effective dynamical equation for a single representative agent. Such an effective equation predicts three long-term dynamic regimes: mono-stability, bi-stability, and non-equilibrium.

Let $s$ denote the strategy played by the representative agent of the population: $s = 1$ if the agent cooperated and $s = 0$ otherwise. According to the EWAN rules, in the mono-stable regime, the probability for a representative agent to cooperate at a generic round, given that at the previous round the agent played strategy $s$ and $n$ of her neighbours cooperated, is given by (see Appendix E in [10]).

$$P_{\text{EWAN}}(C|s, n) = \frac{1}{1 + y_1^{1-\alpha} e^{-\beta \Delta U(s,n)}}, \tag{5}$$

where $y_1 = (1 - x_1)/x_1$, with $x_1$ the only fixed point, $\Delta U(s, n) = (as + b)n + 2hs - h$, and $(a, b, h, \alpha, \beta)$ are parameters defining the mean field dynamics of the model.

When $\beta$ is small we obtain

$$P_{\text{EWAN}}(C|s, n) = m_s n/K + r_s, \tag{6}$$

where $K$ is the number of neighbours in the graph,

$$m_s = \beta K J(as + b), \tag{7}$$

$$r_s = I + \beta J h(2s - 1), \tag{8}$$

and

$$I \equiv \frac{1}{1 + y_1^{1-\alpha}}, \tag{9}$$

$$J \equiv \frac{y_1^{1-\alpha}}{\left(1 + y_1^{1-\alpha}\right)^2}. \tag{10}$$

On the other hand, the MCC rule defined in Section 2 can be written as

$$P_{\mathrm{MCC}}(C|s, n) = \min[1, \, s(p\,n + r) + (1-s)q]. \tag{11}$$

So, in the linear regime, i.e. when the min function is not saturated, we have

$$\tilde{p} = Kp \quad \sim \quad \beta KJ(a+b), \tag{12}$$

$$q \quad \sim \quad I - \beta Jh, \tag{13}$$

$$r \quad \sim \quad I + \beta Jh, \tag{14}$$

and the slope after defection should be vanishing, i.e., $\beta KJb \approx 0$. Notice that $\tilde{p}$ is the slope when we work with the density of neighbouring cooperators, rather than the actual number. This is relevant for heterogeneous systems where the number of neighbours varies. If we want to focus exclusively on homogeneous systems, then we can just compute $p = \tilde{p}/K$, where $K$ is the total number of neighbours. It is important to notice that this estimation is done under the assumption that the min function in the MCC never saturates. If this assumption is not valid, we need to do a more careful estimation.

Eqs (12)–(14) connect the MCC parameters with the more cognitively based parameters of the EWAN model. In Ref. [10] it was shown that, in a mean field approximation, the EWAN model predicted that human groups playing a Prisoner Dilemma Game in Zaragoza [19] are poised near a critical point. Whereas the authors of [10] did not analyze the Madrid experiment, in our analysis here, our results presented in the main text suggest such critical behaviour can be quite common also in humans.

## Supporting information

**S1 Data.**
(PDF)

## Author Contributions

**Conceptualization:** John Realpe-Gómez, Giulia Andrighetto.

**Formal analysis:** Daniele Vilone, John Realpe-Gómez, Giulia Andrighetto.

**Software:** Daniele Vilone.

**Writing – original draft:** Daniele Vilone, John Realpe-Gómez, Giulia Andrighetto.

**Writing – review & editing:** Daniele Vilone, John Realpe-Gómez, Giulia Andrighetto.

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
