## [Decision Letter · Decision Letter 0]

1 Dec 2020

PONE-D-20-34857

Evolutionary advantages of turning points in human cooperative behaviour

PLOS ONE

Dear Dr. Vilone,

Thank you for submitting your manuscript to PLOS ONE. After careful consideration, we feel that it has merit but does not fully meet PLOS ONE’s publication criteria as it currently stands. Therefore, we invite you to submit a revised version of the manuscript that addresses the points raised during the review process.

The three reviewers are in agreement that the paper has good merit and would make a good contribution to the literature of behavioural evolution. They have also suggested several important issues that requrie addressing. It is very important that these issues are seriously considered when the authors prepare the revised version.

We look forward to receiving your revised manuscript.

Kind regards,

The Anh Han, Ph.D.

Academic Editor

PLOS ONE

Additional Editor Comments:

The three reviewers are in agreement that the paper has good merit and would make a good contribution to the literature of behavioural evolution. They have also suggested several important issues that requrie addressing. It is very important that these issues are seriously considered when the authors prepare the revised version.

Journal Requirements:

Reviewers' comments:

Reviewer's Responses to Questions

**Comments to the Author**

1. Is the manuscript technically sound, and do the data support the conclusions?

Reviewer #1: Yes

Reviewer #2: Yes

Reviewer #3: Yes

2. Has the statistical analysis been performed appropriately and rigorously? 

Reviewer #1: Yes

Reviewer #2: Yes

Reviewer #3: Yes

3. Have the authors made all data underlying the findings in their manuscript fully available?

Reviewer #1: Yes

Reviewer #2: Yes

Reviewer #3: Yes

4. Is the manuscript presented in an intelligible fashion and written in standard English?

Reviewer #1: Yes

Reviewer #2: Yes

Reviewer #3: Yes

5. Review Comments to the Author

Reviewer #1: In this work, the authors study, mainly through agent-based simulations, the possibility that the specific behavior of Moody Conditional Cooperators (MCC) that emerge and resist the invasion of free riders in a previous experiment is such that is close to criticality. The link of the evolution of cooperation and criticality is an interesting question to explore and can open the door to new intriguing scientific and even philosophical questions. Therefore I would like to recommend the publication of this work once a few issues are addressed:

1) In the Introduction and Discussion, it is stressed that systems are benefited from being close to criticality. I would recommend to explain a bit more how criticality is defined in this context. For example, being close to criticality means that the system is not finding a stable point? When the system is "too stable" it may favor maladaptive behaviors but it also can reach a good stable point that cannot be reached if it is "oscillating" easily. An explanation and/or discussion clarifying this kind of questions would improve the understanding of the claims.

2) The current work is based in results from one experiment. I recommend to discuss its general applicability (ideally using data or results from other works).

3) Authors fixed the parameters q and r to the values of the experiment and leave the parameter p free in some of their analysis. Why not to do the same with the two first ones? A deeper explanation and/or discussion about the meaning of the parameter p and why it is the key parameter would be advisable.

4) Is p* influenced by the number of All-C and All-D players and then by the initial conditions?

5) The simulation setup shows a few little differences from the experiment. For instance, the number of rounds. Do the authors think that any of these differences could influence p, q, and r?

6) The reader could benefit from a bit more elaborate explanation of the meaning of \\xi.

7) One conclusion that is extracted from this work (and the experiment) is that people cooperate more if more individuals around them did so (it is almost in the core of the definition of MCC). However it has been shown that in repeated collective games sometimes that is not the case. For example, Martinez-Vaquero et al (2020) show that individuals that cooperates only when not enough individuals did so in the previous round to reach a given threshold emerge for given conditions. Could this behavior been influenced by the type of game (weak prisoner's dilemma)? A bit of discussion in this line could be interesting.

8) It would be really interesting if it is possible to include some discussion relating these results (relationship of criticality and cooperation) with other studies in other fields (e.g, sociology, history, sustainability science, etc.)

Reviewer #2: The article if clear and well written. It provides a convincing model and explanation for an experimental observation about the human behavior in cooperation. The results show how cooperative subjects choose or are selected for the minimum level of cooperation able to invade the system in the absence of stubborn defectors, yet being able to resist them in a network.

The reference are adeguate. I think that the article deserves highlighting.

Reviewer #3: Emergence of human collaboration and its evolutionary origins are still a mystery for modern science. Why is cooperation so widespread, when evolution clearly dictates that selfishness is in many ways a more advantages approach? In this paper, authors tackle this evergreen problem by linking the observed properties of a cooperative game-theoretic system to the concept of criticality, drawing parallels between game theory and statistical physics. Combining numerical modeling proposed in reference [1] and experimental results from reference [16], paper shows that indeed the state of the system in which cooperation thrives is reminiscent of a critical state. More precisely, when the system is near such turning point, the population seems to have several evolutionary advantages, including resistance to invasions by free riders.

The findings presented in this paper follow the stream of theory of cooperative games, examined via lens of equilibrium processes, which have been a successful research field for several decades. The paper is clearly written, the conclusions are supported by the results, figures are illustrative and informative. Overall, I am in favor of publishing this paper, as I am convinced it will add value to this research community. However, before my definite acceptance recommendation, I wish that below remarks are addressed.

1. Authors propose a model of prisoner's dilemma game from [1] on 2D lattice and tune the parameters in the way to reproduce certain features observed in the actual experiment with human players reported in [16]. It might be Illustrative for a reader to see this parallel more clearly, for example, by showing some theoretical model curves fitted with experimental data from [16] (such as Fig.1). This work can be seen as an important theoretical re-examination of [16], but I found that obscure reading the paper.

2. In this context, authors should distance their work clearly from both references [1] and [16]. Value added by this paper above that of [1] and [16] should be unquestionable and clearly stated already in Introduction.

3. On the technical side, authors consider agents / players playing a game on 2D lattice. What happens if this is generalized to a lattice with more than two dimensions? This could actually facilitate observations of criticality in the system, enabling the authors to study its role more closely. To clarify, I am not asking for new numerical simulations, but for authors to at least discuss this point. Wouldn’t be interesting to make a setting where closeness to criticality is most easily studied?

4. What always bothers me in studies of this sort is the following. Criticality, traditionally studied in the context of phase transitions and statistical physics, describes a system between two phases, such as between liquid and solid, e.g. water and ice. It is not immediately obvious how this concept can be translated to social systems. Would a common sociologist agree on such interpretation of phases and the order parameter? This of course cannot be answered here, but authors should at least make a stronger argument why do they expect that being near a critical point would be favorable to a social system (assuming we can perfectly well define what such a point means)? I think this is the core theoretical argument behind the whole paper.

5. Literature review is quite comprehensive and complete, yet authors may find it of interest to add below two references:

- Guazzini, F. Stefanelli, E. Imbimbo, D. Vilone, F. Bagnoli, Z. Levnajić, Humans best judge how much to cooperate when facing hard problems in large groups, Scientific Reports 9, 5497, 2019.

- A. Guazzini, D. Vilone, C. Donati, A. Nardi, Z. Levnajić, Modeling crowdsourcing as collective problem solving, Scientific Reports 5, 16557, 2015.

6. PLOS authors have the option to publish the peer review history of their article (what does this mean?). If published, this will include your full peer review and any attached files.

Reviewer #1: No

Reviewer #2: No

Reviewer #3: No

---

## [Author Response · Author response to Decision Letter 0]

8 Jan 2021

We have addressed all the referees' points in the attached response to reviewers.

---

## [Decision Letter · Decision Letter 1]

18 Jan 2021

Evolutionary advantages of turning points in human cooperative behaviour

PONE-D-20-34857R1

Dear Dr. Vilone,

We’re pleased to inform you that your manuscript has been judged scientifically suitable for publication and will be formally accepted for publication once it meets all outstanding technical requirements.

Kind regards,

The Anh Han, Ph.D.

Academic Editor

PLOS ONE

Additional Editor Comments (optional):

Reviewers' comments:

Reviewer's Responses to Questions

**Comments to the Author**

1. If the authors have adequately addressed your comments raised in a previous round of review and you feel that this manuscript is now acceptable for publication, you may indicate that here to bypass the “Comments to the Author” section, enter your conflict of interest statement in the “Confidential to Editor” section, and submit your "Accept" recommendation.

Reviewer #1: All comments have been addressed

Reviewer #3: All comments have been addressed

2. Is the manuscript technically sound, and do the data support the conclusions?

Reviewer #1: Yes

Reviewer #3: Yes

3. Has the statistical analysis been performed appropriately and rigorously? 

Reviewer #1: (No Response)

Reviewer #3: Yes

4. Have the authors made all data underlying the findings in their manuscript fully available?

Reviewer #1: (No Response)

Reviewer #3: Yes

5. Is the manuscript presented in an intelligible fashion and written in standard English?

Reviewer #1: Yes

Reviewer #3: Yes

6. Review Comments to the Author

Reviewer #1: I would like to appreciate authors' time and effort to address reviewers' comments. I believe that the current version of the manuscript has reached the necessary quality level to be published in PLoS ONE.

Reviewer #3: Authors have addressed all my concerns to satisfaction, and I have no further requests from them. I find that the manuscript now is suitable for publication. I congratulate the authors!

7. PLOS authors have the option to publish the peer review history of their article (what does this mean?). If published, this will include your full peer review and any attached files.

Reviewer #1: No

Reviewer #3: No

---

## [Editor Report · Acceptance letter]

27 Jan 2021

PONE-D-20-34857R1

Evolutionary advantages of turning points in human cooperative behaviour 

Dear Dr. Vilone:

I'm pleased to inform you that your manuscript has been deemed suitable for publication in PLOS ONE. Congratulations! Your manuscript is now with our production department.

Kind regards,

on behalf of

Dr. The Anh Han

Academic Editor

PLOS ONE